# Stargardt-like Clinical Characteristics and Disease Course Associated with Variants in the *WDR19* Gene

**DOI:** 10.3390/genes14020291

**Published:** 2023-01-22

**Authors:** Jana Sajovic, Andrej Meglič, Marija Volk, Aleš Maver, Martina Jarc-Vidmar, Marko Hawlina, Ana Fakin

**Affiliations:** 1Eye Hospital, University Medical Centre Ljubljana, Grablovičeva 46, 1000 Ljubljana, Slovenia; 2Clinical Institute of Genomic Medicine, University Medical Centre Ljubljana, Šlajmerjeva 4, 1000 Ljubljana, Slovenia; 3Faculty of Medicine, University of Ljubljana, Vrazov trg 2, 1000 Ljubljana, Slovenia

**Keywords:** *WDR19*, *IFT144*, Stargardt disease, Stargardt-like disease, fundus flavimaculaus, *ABCA4*, phenocopy

## Abstract

Variants in *WDR19 (IFT144)* have been implicated as another possible cause of Stargardt disease. The purpose of this study was to compare longitudinal multimodal imaging of a WDR19-Stargardt patient, harboring p.(Ser485Ile) and a novel c.(3183+1_3184-1)_(3261+1_3262-1)del variant, with 43 ABCA4-Stargardt patients. Age at onset, visual acuity, Ishihara color vision, color fundus, fundus autofluorescence (FAF), spectral-domain optical coherence tomography (OCT) images, microperimetry and electroretinography (ERG) were evaluated. First symptom of WDR19 patient was nyctalopia at the age of 5 years. After the age of 18 years, OCT showed hyper-reflectivity at the level of the external limiting membrane/outer nuclear layer. There was abnormal cone and rod photoreceptor function on ERG. Widespread fundus flecks appeared, followed by perifoveal photoreceptor atrophy. Fovea and peripapillary retina remained preserved until the latest exam at 25 years of age. ABCA4 patients had median age of onset at 16 (range 5–60) years and mostly displayed typical Stargardt triad. A total of 19% had foveal sparing. In comparison to ABCA4 patients, the WDR19 patient had a relatively large foveal preservation and severe rod photoreceptor impairment; however, it was still within the ABCA4 disease spectrum. Addition of *WDR19* in the group of genes producing phenocopies of Stargardt disease underlines the importance of genetic testing and may help to understand its pathogenesis.

## 1. Introduction

Stargardt disease (STGD1, OMIM# 248200), also known as fundus flavimaculatus or ABCA4-retinopathy, is the most frequent retinal dystrophy caused by a single gene, characterized by a progressive degeneration of the retinal pigment epithelium (RPE) and photoreceptors [1,2,3]. It is classically caused by bi-allelic variants in the *ABCA4* gene, localized on chromosome 1p22.1. *ABCA4* gene encodes a transmembrane protein ABCA4, which is present in outer segments of photoreceptors and inner membranes of RPE [4,5,6]. It transports molecules involved in visual transduction, thus being an essential part in the visual cycle, and involved in removal of toxic vitamin A products [7]. Stargardt disease most commonly begins in childhood or adolescence and, although it is known for a very heterogeneous phenotypic appearance, typical Stargardt triad of central atrophy of RPE/photoreceptors, flecks, and peripapillary sparing is found in most patients [1,3,8]. Fundus appearance is usually classified according to Fishman groups: I—flecks limited to within the vascular arcades, II—fleck-like lesions anterior to the vascular arcades and/or nasal to the optic disc, III—most diffuse flecks resorbed leaving diffuse RPE atrophy, and IV—not only diffusely resorbed fundus flecks and atrophy of the RPE but also diffuse choriocapillaris atrophy. Based on electroretinography (ERG), patients can have either normal peripheral retinal function (ERG group 1) or develop cone/cone-rod dystrophy (ERG group 2 and 3, respectively) [9]. Fovea is typically affected early in the course of disease; however, approximately 20% of patients have a subtype of disease with delayed foveal involvement, i.e., “foveal sparing” [10,11,12,13].

Several other genes have been associated with clinical presentation similar to ABCA4-retinopathy and are important to consider in differential diagnosis. The most frequent phenocopies are caused by variants in *ELOVL4, PROM1*, and *PRPH2* [14,15,16,17,18]. *ELOVL4* is located on chromosome 6q14.1 and encodes a protein, present in the endoplasmic reticulum of photoreceptors [19]. It is involved in the biosynthesis of very-long-chain fatty acids (VLC-PUFA) that are enriched in the retina [20]. Stargardt-like phenotype caused by variants in *ELOVL4* has been labelled Stargardt type 3 (STGD3, OMIM# 600110) [14,15,16,17]. *PROM1* is located on chromosome 4p15.32 and encodes prominin 1, present at the base of the outer segments of photoreceptors, where it plays an important role in the disc morphogenesis, and RPE, where it regulates autophagy [21,22]. Stargardt-like phenotype caused by variants in *PROM1* has been labelled Stargardt type 4 (STGD4, OMIM# 603786) [17,18,23,24,25]. *PRPH2* is located on chromosome 6p21.1 and encodes peripherin 2, present in outer segments of photoreceptors, where it is involved in the morphogenesis, stabilization, and compaction of outer segment discs and lamellae [26,27]. Stargardt-like phenotype caused by variants in *PRPH2* has been labelled multifocal pattern dystrophy simulating fundus flavimaculatus (PDSFF) [28,29]. Other genes that may cause retinal disease with characteristics of Stargardt disease are *CRX*, *BEST1*, *CDH3*, *CERKL*, *RDH12*, *RPGR*, and *IMPG1* [1,30,31,32]; however, without all characteristics of the typical Stargardt triad. The term STGD2 was eventually discarded, as it was shown to be caused by the same gene as in STGD3 [19].

Recently another gene, *WDR19* (also known as *IFT144* or *NPHP13*), has been implicated in causing Stargardt-like phenotype [33,34]. *WDR19* is located on chromosome 4p14 and encodes a ciliary protein named intraflagellar transport 144 protein (IFT144) [35]. *WDR19* has been classically associated with ciliopathies involving retinitis pigmentosa (RP), such as Sensenbrenner, Jeune syndrome, [36], nephronophthisis [35], Caroli disease [37] and Senior-Løken syndrome [38]. In 2017, Stone et al. were the first to report an association between *WDR19* and Stargardt-like phenotype [34]. In 2020, Shamseldin et al. published a report of three unrelated patients with a founder mutation [33]. These reports did not contain any clinical data with the exception of one color fundus image [33].

The purpose of this study was to extend the clinical description of WDR19-Stargardt phenotype using state-of-the-art methodology, including fundus autofluorescence (FAF), spectral-domain optical coherence tomography (OCT) and ERG. Furthermore, the phenotype was compared with that of ABCA4 patients, exhibiting the prototypical Stargardt phenotype.

## 2. Materials and Methods

The study included a 25-year old male patient with Stargardt-like disease, harboring two *WDR19* variants (WDR19 patient), and 43 patients (13 male, 30 female; median age 38 years, range 11–75 years) with Stargardt disease, harboring *ABCA4* variants (i.e., ABCA4-retinopathy) (ABCA4 patients). The patients were recruited from the Slovenian registry of 1157 patients with inherited retinal diseases. Clinical characteristics (defined in detail below) were compared between a WDR19 patient and ABCA4 patients. The ABCA4 patients were sub-stratified based on genotype (double null or other) and preservation of the fovea (foveal sparing or not). For the purpose of this study, foveal sparing was defined as foveal preservation on FAF and OCT, with or without surrounding RPE atrophy [11]. As disease was mostly symmetrical between the two eyes, the right eye was chosen for analysis.

The study was conducted according to the guidelines of the Declaration of Helsinki. Study was reviewed and approved by the National medical ethics committee of the Republic of Slovenia (protocol ID number: 0120-50/2021/3). All examinations were completed as a part of routine diagnostic procedures. Written informed consent was obtained from participants before their enrolment.

### 2.1. Genetic Analysis

The patient harboring *WDR19* variants underwent genetic testing using a buccal swab sample and targeted genetic screening of 351 genes, including *ABCA4*, *ELOVL4*, *PROM1* and *PRPH2.* Blueprint Genetics Retinal Dystrophy Panel (test code OP0801) was used. Exome sequencing was performed using a custom target capture approach, which targets coding regions of genes, mitochondrial genome and also includes those non-coding regions where pathogenic non-coding variants were previously reported in the literature and databases. Other non-coding regions, including 5′UTR, are not captured in this approach. Of ABCA4 patients, 21 of them underwent genetic testing with the panel Retinal disorders v2.14 (list is available on request) that included the *WDR19* gene (none of those had variants in the *WDR19* gene), while 22 patients had only *ABCA4* gene analysis.

### 2.2. Clinical Analysis

All patients underwent a detailed ophthalmological exam. Best-corrected visual acuity (BCVA) was measured using Snellen charts. In some patients, at their last visit, Snellen vision was determined additionally by using Tabletop Refraction System TS-610 (Nidek Co., Ltd., Gamagōri, Japan). Color vision was measured using Ishihara tables. Visual field was examined with static perimetry using an Octopus 900 (Haag-Streit International, Koeniz, Switzerland). Age at onset was defined as the age at which patients first noted decreased visual acuity (VA). Disease duration was calculated as the time between the age at the last exam and the age at onset. Slit lamp examination, color fundus imaging, FAF imaging and OCT imaging were executed after pupil dilation with topical 1% tropiciamide. Color fundus image was taken with Topcon retinal camera TRC-50DX (Topcon Corporation, Tokyo, Japan). 55° or 30° FAF imaging was performed using Heidelberg Spectralis (Heidelberg Engineering, Heidelberg, Germany) and ultra-wide field FAF using Panoramic Ophthalmoscope P200DTx (Optos plc, Dunefermline, United Kingdom). OCT macular scans were performed using the Spectralis OCT+HRA device (Heidelberg Engineering, Heidelberg, Germany).

From the 30° FAF images, areas of definitely decreased autofluorescence (DDAF), representing RPE atrophy, were measured using our custom code written in MATLAB (The MathWorks, Inc., Natick, MA, USA). According to ProgStar criteria, DDAF was defined as being at least 90% black in reference to optic nerve / main blood vessels (100%). The opposite reference point represented a healthy retina (0%). Optic nerve and blood vessels were excluded. The analysis of DDAF included a subset of 18 ABCA4 patients from another study (harboring either two null variants or a combination of c.5714+5G>A; (p.[=,Glu1863Leufs * 33]) and a null variant). The WDR19 patient had unusually hypoautofluorescent foveal region that was, based on OCT findings, normally preserved and was therefore excluded from the DDAF area measurement.

Pattern ERG (PERG) and multifocal ERG (mfERG) were used to evaluate macular function, while scotopic and photopic full-field ERG (ffERG) was used to evaluate generalized retinal function. The recordings were made according to the standards of the International Society of Clinical Electrophysiology of Vision (ISCEV) [39,40,41,42]. The recording electrode was an HK loop placed in the fornix of the lower eyelid [43], silver chloride reference electrode was placed on the ipsilateral temple and the ground electrode was positioned on the forehead.

Retinal sensitivity to light and fixation were evaluated in WDR19 patient in mesopic conditions with a Nidek MP1 microperimeter (Nidek Technologies, Padova, Italy) after pupil dilation with topical 1% tropiciamide. Adjusted 10-2 Humphrey test grid was used, comprising 56 test loci, covering 20 × 20°. The test stimulus was white and was set to Goldmann size III. The stimulus intensity varied from 127 to 1.27 cd/m^2^, corresponding to retinal sensitivities of 0 dB to 20 dB. A 4-2 threshold strategy was used. Sensitivity values from all 56 test loci and fixation were then superimposed over a 55° FAF image using the MP1 microperimeter software (NAVIS software version 1.7.9, Nidek Technologies, Padova, Italy).

## 3. Results

### 3.1. Molecular Results

Sequence analysis identified two variants of uncertain significance (VUS) in the *WDR19* gene (reference transcript NM_025132.4): c.1454G>T; p.(Ser485Ile) and c.(3183+1_3184-1)_(3261+1_3262-1)del; p.(?). The first is a missense variant, which is not found in gnomAD genomes or exomes [PM2]. However, this variant has been reported in a compound heterozygous state with c.1031A>G (p.His344Arg) in a family with two affected individuals with Stargardt-like phenotype [34] [PM3_SUP]. The second one is a novel variant and is also not found in gnomAD genomes nor exomes [PM2]. It results in the in-frame deletion of exon 29 of the *WDR19* gene [PVS1_MOD]. Variants were classified according to ACMG guidelines for the interpretation of sequence variants [44,45]. Both variants are classified as VUS, as there is insufficient evidence to conclusively assert their pathogenicity. The patient had no identified pathogenic variants in *ABCA4*, *ELOVL4*, *PROM1* or *PRPH2.* There were no other affected family members.

### 3.2. Review of Known WDR19 Variants and Their Associated Phenotypes

The list of previously identified *WDR19* variants are shown in Appendix A. Stargardt phenotype has been noted in four patients harboring *WDR19* variants, notably, c.1031A>G, c.1454G>T, c.2777G>T [33,34].

### 3.3. Clinical Data

The onset of visual symptoms of the WDR19 patient was approximately 5 years of age, when parents noticed that he had problems seeing at night (e.g., being cautious when walking in poorly lit environments). At that time VA was 0,8 Snellen decimal and an irregular foveal reflex was noted; however, color fundus imaging did not show any abnormalities (Figure 1A). Over the years, the patient noticed increasing problems seeing at night or in a dim light and at the age of 18 years, a detailed ophthalmological exam was performed. At that time, VA and color vision were normal and fundus exam was within normal limits. However, FAF showed a few hyperautofluorescent flecks and OCT showed hyper-reflectivity at the level of the external limiting membrane (ELM)/outer nuclear layer (ONL) (Figure 2). Over the follow-up of 7 years, VA and color vision remained preserved, while extensive yellow flecks appeared on the posterior pole, extending outside the vascular arcades (Fishman II). FAF showed increasing number of hyperfluorescent perifoveal flecks extending from the perifovea to outside of the vascular arcades, later transforming to spots of hypoautofluorescence (progressing from Fishman 0 to Fishman III) (Figure 1A–E). Perifoveal loss of photoreceptors appeared on OCT at the age of 24 years. Peripapillary and foveal regions were still relatively well preserved at the last imaging at aged 25 years. At the ages of 24 and 25, static assessment of the central 60° visual field showed reduced pericentral sensitivity, which corelated nicely with the photoreceptor atrophy. Microperimetry, performed at the ages of 20 and 24 years, showed loss of retinal sensitivity in the perifoveal area (Figure 2).

ERG performed at the age of 19 years revealed generalized photoreceptor dysfunction (ERG group 3). Function of cones and rods was severely decreased and the amplitudes reflecting rod responses being barely detectable above the noise level. MfERG showed preserved function in the central foveal ring. The follow-up ERG exam at the age of 24 years, revealed a decline in all responses, while the foveal function still remained relatively preserved. The ERG responses including PERG P50 amplitude, mfERG, dark-adapted (DA) 0.01 ERG b-wave, DA 3.0 ERG a-wave, oscillatory potentials (OP), light-adapted (LA) 30 Hz flicker ERG, LA 3.0 ERG b-wave and S-cone ERG from the last ERG exam are shown in Figure 3. The patient was otherwise healthy and had no renal impairment or other diseases and malformations.

### 3.4. Comparison between WDR19 and ABCA4 Stargardt Patients

Since the WDR19 patient presented with a Stargardt disease phenotype, a comparison with 43 ABCA4-Stargardt patients was performed. The main phenotypic characteristics are shown in Table 1.

A total of 8 out of 43 (19%) ABCA4 patients had preserved fovea on the initial OCT imaging (marked with empty circles on Figures 5 and 7–12). FAF and OCT of five representative cases are shown in Figure 4 for comparison with the WDR19 patient. The WDR19 patient had a relatively large extent of foveal preservation. Two of the ABCA4 patients had no signs of flecks or parafoveal RPE atrophy at the first imaging (Figure 4B,C). One of them (Figure 4B, harboring two null variants) lost foveal photoreceptors after 2 years, while they were still preserved at the latest follow-up (after four years) in the second patient (Figure 4C). Hyper-reflectivity at the level of ELM/ONL that was present the WDR19 patient was also observed in the two ABCA4 patients.

Associations between VA and age for WDR19 and ABCA4 patients are shown in Figure 5. Preserved VA of the WDR19 patient was similar to that of foveal sparing ABCA4 patients. Associations between RPE atrophy (represented by DDAF area) and age for WDR19 and ABCA4 patients are shown in Figure 6. WDR19 patient had DDAF area in the range of ABCA4 group, in the size similar to non-null (other) *ABCA4* genotypes. Correlations between different ERG amplitudes and age for WDR19 and ABCA4 patients are shown in Figure 7, Figure 8, Figure 9 and Figure 10. ERG amplitudes of WDR19 patient were within the range observed in the ABCA4 patient group, with the ffERG responses resembling double null ABCA4 patients (Figure 7, Figure 8 and Figure 9) and PERG P50 resembling other *ABCA4* genotypes (Figure 10). For patients with different genotypes, additional boxplot charts showing VA, DDAF area, DA 0.01 ERG b-wave amplitude, DA 3.0 ERG a-wave amplitude, LA 30 Hz ERG amplitude and PERG P50 amplitude are presented in a Appendix A.

To evaluate whether there are any specific characteristics of the WDR19-Stargardt phenotype, analyses of parameters reflecting the ratios of rods vs. cone impairment, macular vs. peripheral retinal impairment, and photoreceptor vs. RPE impairment were performed. The relationship between the rod and cone photoreceptor impairment was studied by comparing DA 0.01 ERG (reflecting rod system function) and LA 30 Hz ERG (reflecting cone system function) amplitudes (Figure 11). The WDR19 patient had relatively lower rod responses in comparison to ABCA4 patients regardless of genotype, but still within the group’s variability. The relationship between macular and peripheral retinal involvement was studied by comparing PERG P50 (reflecting macular function) vs. LA 30 Hz ERG (reflecting mostly peripheral retinal function) amplitudes (Figure 12). The WDR19 patient had relatively higher PERG P50 amplitude in comparison to ABCA4 patients but still within the group’s variability. The relationship between photoreceptor and RPE impairment in the macula was studied by comparing DDAF area (reflecting RPE atrophy) and PERG P50 (reflecting macular photoreceptor function) amplitude (Figure 13). The WDR19 patient had values within the ABCA4 group’s variability, in similarity closer to the “other” ABCA4 group.

## 4. Discussion

The study reports a detailed phenotypic analysis of a WDR19 patient with a Stagart-like phenotype. The patients’ phenotype was by and large undistinguishable from ABCA4-Stargardt disease; however, there were some atypical findings, notably the degree of foveal sparing and the relatively severe involvement of rod photoreceptors.

### 4.1. Genetic Considerations

The presented patient harboured a previously reported heterozygous missense variant p.(Ser485Ile) and a novel heterozygous deletion c.(3183+1_3184-1)_(3261+1_3262-1)del, encompassing exon 29 of the *WDR19* gene. Single or multi exon heterozygous deletions in the *WDR19* gene have previously been reported in patients with WDR19-related conditions [46]. To date, no patients with WDR19-associated Stargard-like phenotype have been reported to have candidate copy number loss/gain variants. Some pathogenic copy number variants encompassing single or multiexon deletion have been associated with WDR19-related conditions [46] (see also ClinVar database). We added reported copy number variants in the *WDR19* gene in the Appendix A, which sums reported pathogenic variants in the *WDR19* gene. Variant p.(Ser485Ile) has been reported as compound heterozygous with *WDR19* c.1031A>G; p.(His344Arg) in a family with two affected individuals with Stargardt-like phenotype [34]. It is likely that variants in our WDR19 patient are the cause of the disease, as the Stargardt-like phenotype has already been described in association with *WDR19* [33,34] and the patient did not carry pathogenic variants in any other genes causing Stargardt-like disease. Both identified variants were absent from the gnomAD project (these variants do not have gnomAD exomes or genome entry despite good coverages of the loci). In addition, we did not detect these variants in our in-house database (more than 10,000 Slovenian exomes).

Biallelic pathogenic and likely pathogenic variants in the *WDR19* gene present an established cause of autosomal recessive Cranioectodermal dysplasia 4 (OMIM: 614378), Nephronophthisis 13 (OMIM: 614377), Senior-Loken syndrome 8 (OMIM: 616307) and Short-rib thoracic dysplasia 5 with or without polydactyly (OMIM: 614376). There are only two reports of *WDR19* variants associated with Stargardt-like disease. These variants are p.(His344Arg), p.(Ser485Ile) and p.(Ser926Ile) [33,34]. While the first two variants were novel, the letter was reported in trans with another pathogenic variant in association with cranioectodermal dysplasia [47]. It has been hypothesized that more severe *WDR19* pathogenic variants may lead to multisystemic ciliopathies, while hypomorphic variants may be associated with non-syndromic retinal dystrophy (RP, macular dystrophy) [38]. However, other yet unrecognized mechanisms may impact the phenotypic expression of the *WDR19* gene.

### 4.2. Clinical Findings and Comparison with ABCA4-Stargardt Phenotype

Many similarities were noted, when comparing the phenotype of the WDR19 patient with that of ABCA4-Stargardt patients. The WDR19 patient exhibited the typical Stargardt triad of macular atrophy (albeit in the perifoveal region), peripapillary sparing and widespread hyper- and hypoautofluorescent flecks. The fovea remained preserved after perifoveal atrophy occurred, similar to the phenotype observed in >10 % of Stargardt patients, who exhibit the so called foveal sparing phenotype [10,11,48]. The term “foveal sparing” is applied by some authors only to those patients that retain foveal structure next to extensive perifoveal RPE atrophy, and these mostly present with symptoms after the age of 50 years [10,48]. However, less strict definitions have also been used [11], the common denominator being that the patients do not exhibit early-onset foveal atrophy [10,11]. For the purpose of this study, we considered the patients to have foveal sparing if the fovea was preserved on presentation (examples in Figure 4), even though some lost the fovea soon after. The more accurate description of Stargardt phenotypes may be 1) early onset foveal atrophy, 2) delayed foveal atrophy, and 3) foveal sparing. Longitudinal follow-up will reveal whether the fovea of the WDR19 patient remains preserved long-term. Interestingly, the color fundus image of another WDR19-Stargardt patient provided by the Shamseldin et al. also seem to exhibit preservation of the fovea [33]; however, the definitive conclusion could not be made as there was no functional data. The WDR19 patient in the present study also had an unusually hypoautofluorescent foveal region compared to ABCA4 patients; however, it is unclear whether the sign is due to true hypoautofluorescence or only appears dark due to the surrounding hyperautofluorescence.

The WDR19 patient exhibited hyperreflective changes at the level of ELM/ONL, that remained during the whole 7 years of follow-up with OCT imaging (Figure 2), and the same sign was observed in two of the ABCA4 patients (Figure 4). This feature has been previously described in childhood-onset ABCA4 patients as one of the earliest OCT abnormalities [49]. Although the authors suggested that this structural change suggests metabolically stressed photoreceptors before cell death [49], curiously the photoreceptors of the WDR19 patient remained preserved for 7 years, and similarly, one of the two ABCA4 patients (Figure 4C) had this sign and preserved photoreceptors for 4 years.

As ABCA4 genotypic and phenotypic spectrum is extremely diverse [1,3,50,51], we attempted to determine whether the phenotype of the WDR19 patient more closely resembles that of the double null ABCA4 patients or that of ABCA4 patients with non-null genotypes. Interestingly, the WDR19 patient could not fit easily in either group. Relatively good macular preservation (large PERG P50 amplitude, small DDAF, and foveal sparing) more closely resembled the phenotype of non-null ABCA4 patients; however, severe generalized photoreceptor dysfunction on ffERG at a young age more closely resembled the phenotype of double null ABCA4 patients. The rod function was low even when compared to double null ABCA4 patients and the patient reported nyctalopia as the presenting symptom, which is unusual for Stargardt disease. *WDR19* variants are mostly linked to RP (i.e., rod-cone dystrophy) [38]. A study on a larger group of patients is needed to confirm whether WDR19-Stargardt patients exhibit more rod dysfunction than ABCA4-Stargardt patients.

### 4.3. WDR19 and Other Genes Producing Phenocopies of Stargardt Disease

*WDR19,* a ciliary gene, is expressed in photoreceptors at the level of the connecting cilium, and encodes IFT144 protein, which is a component of the IFT-A transport complex. Although IFT-A complex is classically involved in retrograde transport, the studies showed that IFT144 protein is required for the efficient transport of opsins and the distal elongation of outer segments, which suggest its indirect involvement in the anterograde transport [52,53]. WDR19-associated retinal diseases are inherited in an autosomal recessive fashion. Among retinal diseases, it was reported to cause Stargardt-like disease [33,34] and isolated RP [38] or syndromic RP. RP can be restricted to the eye, present together with sub-clinical renal cysts or associated with a syndrome (e.g., Senior Løken and Sensenbrenner syndromes) [35,38]. The WDR19-Stargardt disease has not been associated with a syndromic disease, as was also the case in our patient.

Similar to the *WDR19* gene, other genes that produce Stargardt phenocopies (*ELOVL4, PROM1*, and *PRPH2*) also exhibit phenotypic diversity, often in association with different inheritance patterns. Autosomal dominant mutations in *ELOVL4* gene cause STGD3 [14], while autosomal recessive mutations result in RP [54,55,56,57] and LCA [54,58]. This is also true for the *PROM1* gene, where recessive form resembles RP [23,59,60], while dominant *PROM1* mutations leads to STGD4 and other forms of macular dystrophies [25,30,61]. *PRPH2* gene is linked to the most heterogeneous spectrum of retinal diseases. It has an autosomal dominant pattern of inheritance and specific *PRPH2* variants causes certain phenotypes: Stargardt-like disease/fundus flavimaculatus, retinitis punctata albescens, RP, digenic RP, extensive chorioretinal atrophy, central areolar choroidal dystrophy, retinitis punctata albescens, pattern dystrophy, adult-onset vitelliform macular dystrophy and other unspecified autosomal dominant macular dystrophies [28,29,62,63,64,65]. The addition of *WDR19* to the group of Stargardt disease causing genes may help to understand the pathogenesis of this frequent, yet untreatable disease. The proteins encoded by *ABCA4* and the four above-mentioned genes are not known to be directly linked; however, ABCA4 and PRPH2 and PROM1 occur on a similar location in the outer segments, WDR19 is involved in transport, and ELOVL4 is involved in synthesis of very long chain polyunsaturated fatty acid (VLC-PUFA), implicated in lipofuscin accumulation [66]. We hypothesize that these proteins play a cooperative role in at least one specific pathway, crucial for normal photoreceptor function.

### 4.4. Advantages and Disadvantages of the Study

This study for the first time provided a complete ophthalmological description of a WDR19-Stargardt patient. The advantage of the study was a long follow-up of 20 years and detailed imaging and functional analysis. The study is strengthened by a comparison of the WDR19 phenotype with an equally well documented cohort of patients with ABCA4-Stargardt disease. An obvious disadvantage of the study is that only one WDR19 patient is included, which makes it difficult to address the specific findings that suggested some deviation from the classical Stargardt phenotype.

## 5. Conclusions

The study showed that the WDR19-Stargardt phenotype is within the phenotypic spectrum of ABCA4-Strgardt disease, stressing the importance of genetic testing in patients with inherited retinal dystrophies. The addition of *WDR19* in the group of genes producing Stargardt phenocopies may help to understand the pathogenesis behind this frequent yet untreatable disease.

## Figures and Tables

**Figure 1 genes-14-00291-f001:**
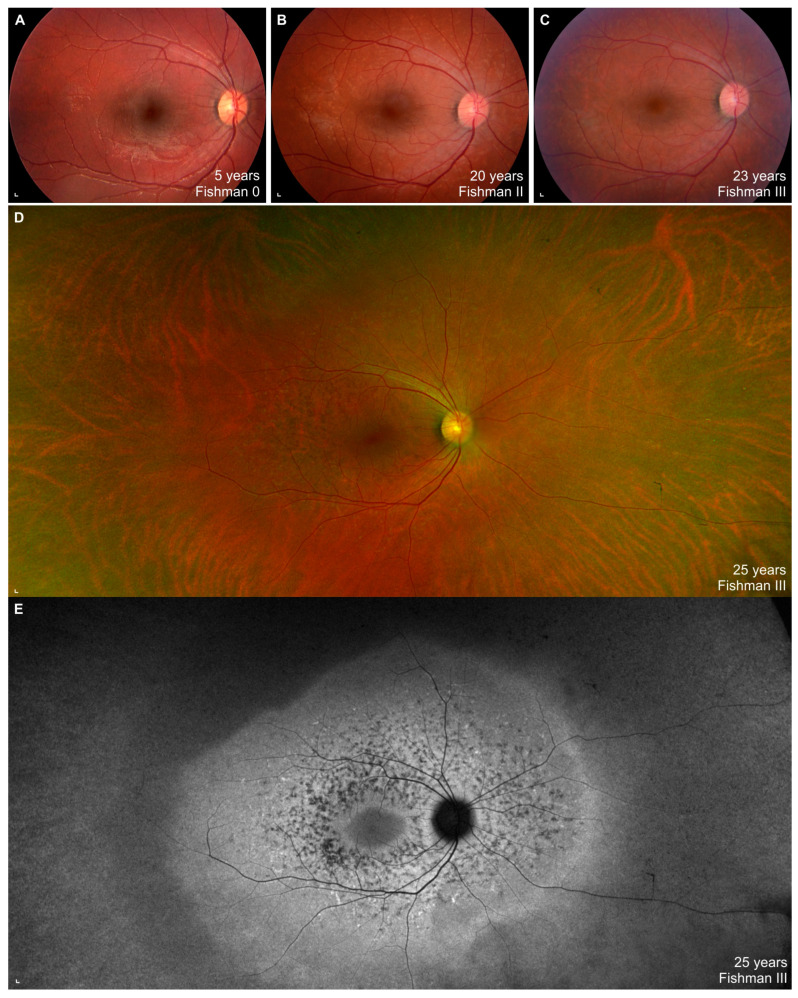
Right eye color fundus images and fundus autofluorescence (FAF) Optos image from a WDR19 patient from the first until the last exam. Note the progression from normal fundus ((**A**); Fishman 0) to extensive flecks ((**B**); Fishman II) to resorbed flecks ((**C**–**E**); Fishman III). Scale bars: 200 µm.

**Figure 2 genes-14-00291-f002:**
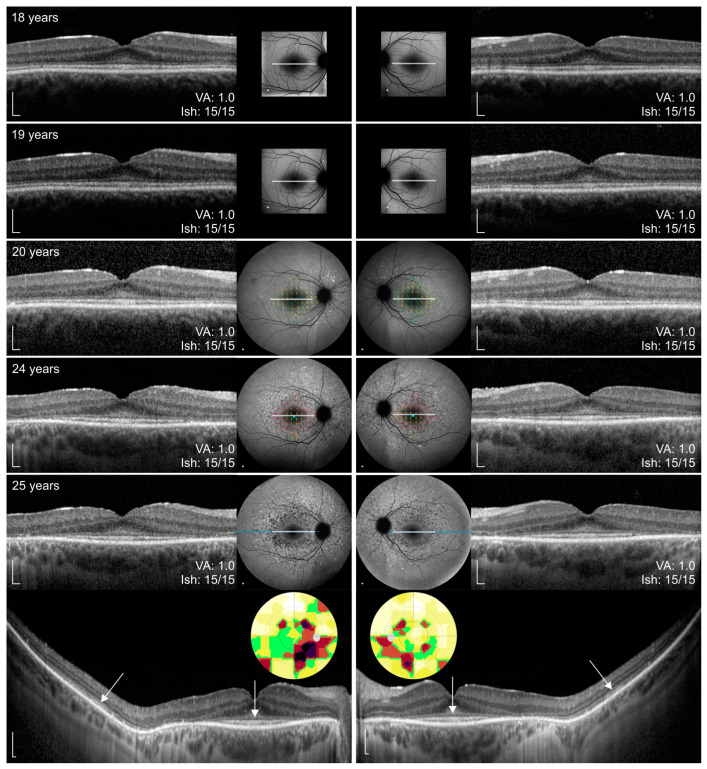
Right eye (**left column**) and left eye (**right column**) spectral-domain optical coherence tomography (OCT) and FAF images of WDR19 patient over the course of 7 years. Age at the exam, best-corrected visual acuity (BCVA) in Snellen, Ishihara color vision are stated on each image. Microperimetry exam is overlaid on FAF images in selected exams. Static perimetry covering central 60° degrees is shown on the bottom. Note the typical signs of Stargardt disease, i.e., peripapillary sparing and hyper-and hypoautofluorescent flecks. Macular atrophy is most notable in the perifoveal area, while fovea is preserved (i.e., foveal sparing). Inner segment ellipsoid (ISe) band is marked with arrows. Abbreviation explanation: VA—visual acuity, Ish—Ishihara plates. Scale bars: 200 µm.

**Figure 3 genes-14-00291-f003:**
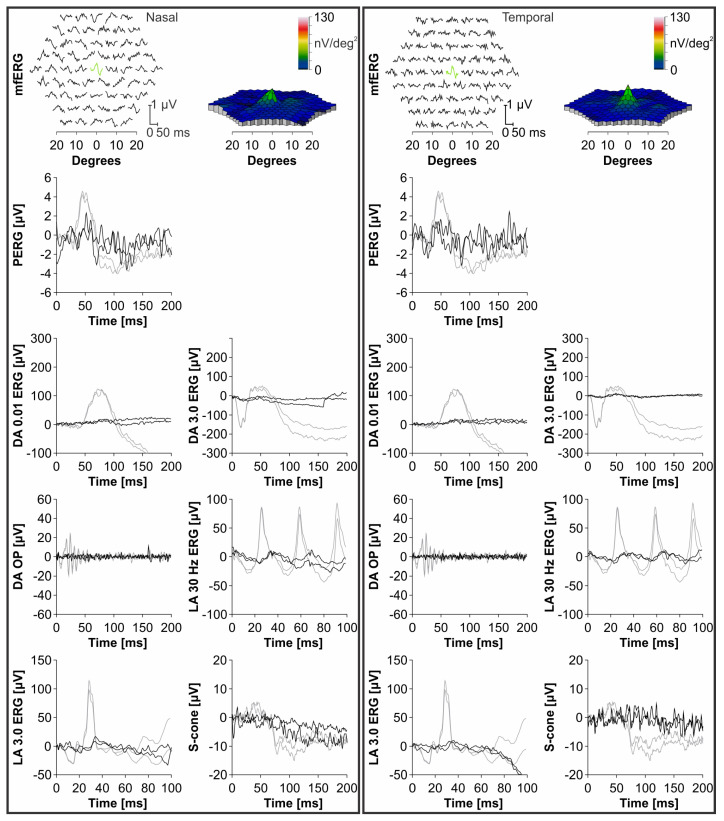
Right eye (**left column**) and left eye (**right column**) electroretinography (ERG) responses of the WDR19 patient at 24 years of age (black lines) and a healthy control (grey lines). Multifocal ERG (mfERG) responses of the WDR19 patient are shown at the top with a 3D representation of the values on the right side. All ERG responses of the WDR19 patient were below the normal range. Note the relatively severe loss of ERG responses representing rod system function (dark-adapted (DA) 0.01 ERG and DA 3.0 ERG) in comparison to light-adapted (LA) 30 Hz, which represents cone system function. Preserved foveal function in the first mfERG ring is marked with green. Abbreviation explanation: PERG—pattern ERG, OP—oscillatory potentials.

**Figure 4 genes-14-00291-f004:**
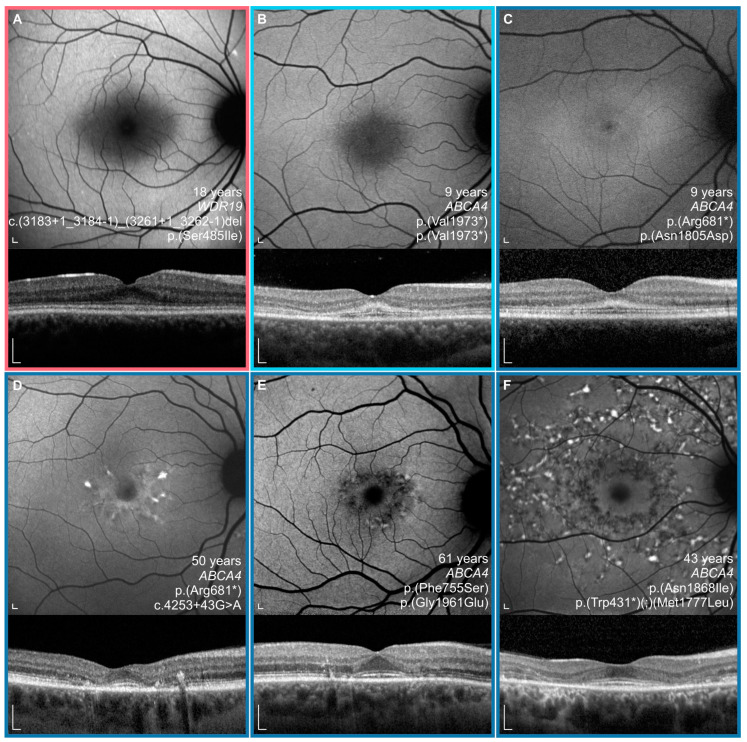
Characteristics of a WDR19 patient (**A**) and ABCA4 patients (**B**–**F**) and with foveal sparing. Note a relatively large extent of foveal preservation in the WDR19 patient. WDR19 patient is encircled with red (**A**), double null ABCA4 patient with light blue (**B**) and ABCA4 patients with other genotypes with dark blue (**C**–**F**). Note the hyper-reflectivity at the level of the external limiting membrane (ELM)/outer nuclear layer (ONL) in the WDR19 patient and two ABCA4 patients (**B**,**C**). Note also the relatively dark fovea of the WDR19 patient and two of the ABCA4 patients (**E**,**F**). Scale bars: 200 µm.

**Figure 5 genes-14-00291-f005:**
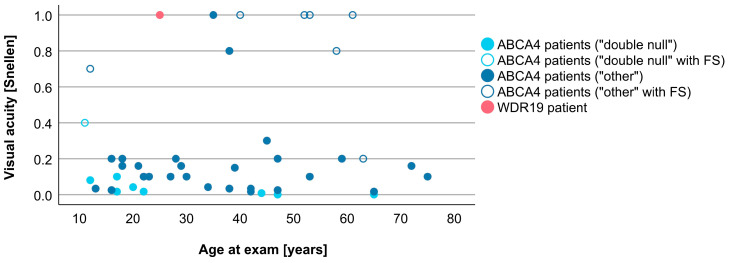
Comparison of VA between WDR19 patient (red), double null ABCA4 patients (light blue) and ABCA4 patients with other genotypes (dark blue). Foveal sparing patients are marked with empty circles. Abbreviation explanation: FS—foveal sparing.

**Figure 6 genes-14-00291-f006:**
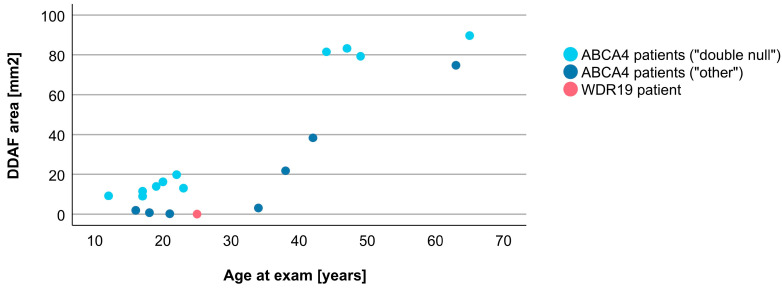
Comparison of DDAF area, representing retinal pigment epithelium (RPE) atrophy, between WDR19 patient (red), double null ABCA4 patients (light blue) and ABCA4 patients with other genotypes (dark blue). For this analysis a limited number of “other” ABCA4 patients (only those harboring *ABCA4* variant c.5714+5G>A (p.[=,Glu1863Leufs*33]) was included (see Section 2).

**Figure 7 genes-14-00291-f007:**
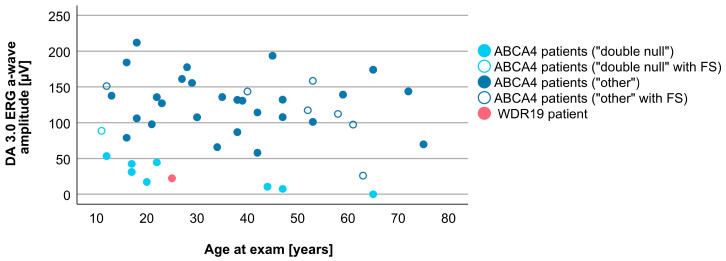
Comparison of DA 3.0 ERG a-wave amplitudes (providing a measure of rod photoreceptor funtion) between WDR19 patient (red), double null ABCA4 patients (light blue) and ABCA4 patients with other genotypes (dark blue). Foveal sparing patients are marked with empty circles.

**Figure 8 genes-14-00291-f008:**
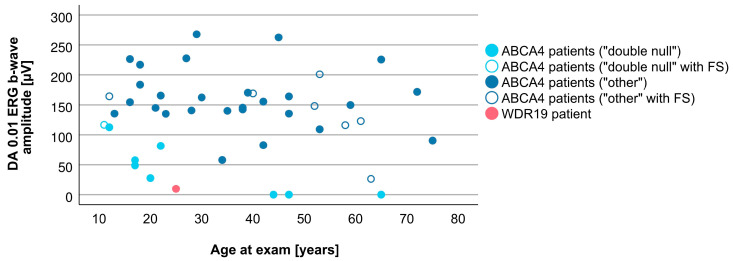
Comparison of DA 0.01 ERG b-wave amplitude (representing rod system function) between WDR19 patient (red), double null ABCA4 patients (light blue) and ABCA4 patients with other genotypes (dark blue). Foveal sparing patients are marked with empty circles.

**Figure 9 genes-14-00291-f009:**
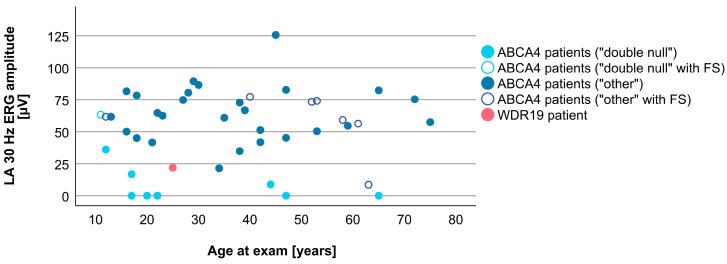
Comparison of LA 30 Hz ERG amplitudes (representing cone system function) between WDR19 patient (red), double null ABCA4 patients (light blue) and ABCA4 patients with other genotypes (dark blue). Foveal sparing patients are marked with empty circles.

**Figure 10 genes-14-00291-f010:**
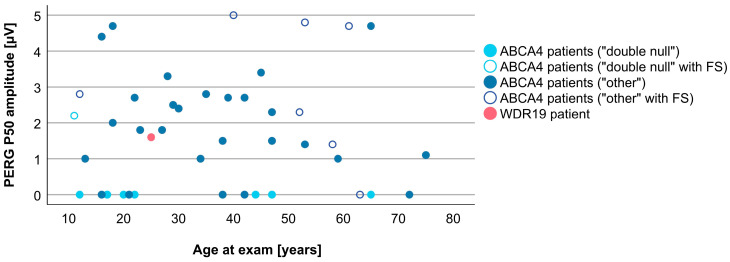
Comparison of PERG P50 amplitudes (representing macular function) between WDR19 patient (red), double null ABCA4 patients (light blue) and ABCA4 patients with other genotypes (dark blue). Foveal sparing patients are marked with empty circles.

**Figure 11 genes-14-00291-f011:**
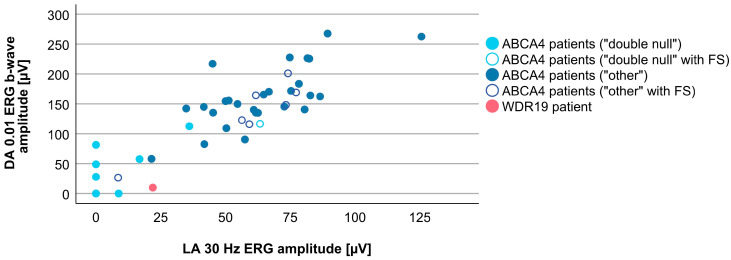
Different ratios of cones and rods impairment in WDR19 patient (red), double null ABCA4 patients (light blue) and ABCA4 patients with other genotypes (dark blue). Foveal sparing patients are marked with empty circles.

**Figure 12 genes-14-00291-f012:**
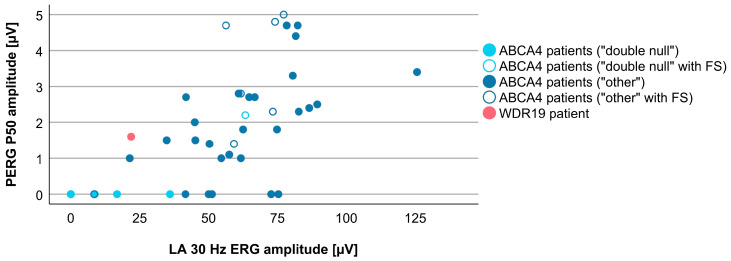
Different ratios of cone system and macular photoreceptors impairment in WDR19 patient (red), double null ABCA4 patients (light blue) and ABCA4 patients with other genotypes (dark blue). Foveal sparing patients are marked with empty circles.

**Figure 13 genes-14-00291-f013:**
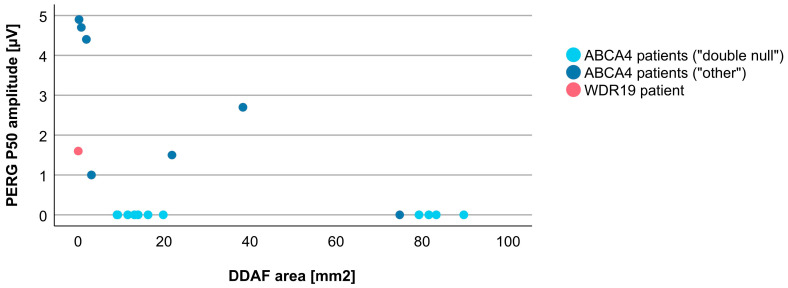
Different ratios of RPE and photoreceptor impairment in WDR19 patient (red), double null ABCA4 patients (light blue) and ABCA4 patients with other genotypes (dark blue). For this analysis a limited number of “other” ABCA4 patients (only those harboring *ABCA4* variant c.5714+5G>A (p.[=,Glu1863Leufs * 33]) was included (see Materials and Methods).

**Table 1 genes-14-00291-t001:** Comparison of clinical characteristics of the WDR19 and ABCA4 Stargardt patients.

Parameter	WDR19 Patient	ABCA4 Patients (Median, Range)
Age at exam	24 years	38 years (11–75 years)
Age at onset	5 years	16 years (5–60 years)
Disease duration	21 years	12 years (0–55 years)
Visual acuity (Snellen decimal)	1.0	0.1 (0.0–1.0)
PERG P50 amplitude	1.6 µV	1.5 µV (0.5–5.0 µV)
S-cone ERG amplitude	0.0 µV	3.4 µV (0.0–8.9 µV)
DA 0.01 ERG b-wave amplitude	9.8 µV	143.6 (0.0–267.8 µV)
DA 3.0 ERG a-wave amplitude	22.3 µV	113.3 µV (0.0–211.9 µV)
Oscillatory potentials	0.0 µV	9.3 µV (0.0–53.9 µV)
LA 30 Hz ERG amplitude	21.9 µV	58.4 µV (0.0–125.6 µV)
LA 3.0 ERG b-wave amplitude	16.7 µV	60.5 µV (0.0–128.7 µV)
DDAF area	0.04 mm^2^	15.09 mm^2^ (0.22–89.67 mm^2^), N = 18

Abbreviation explanation: DDAF—definitely decreased autofluorescence. Values recorded at the last ERG exam were taken for the analysis.

## Data Availability

The data that support the results of this study are available upon request from the corresponding author, A.F. The data are not publicly available due to personal data protection policies.

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
