# Peer review of "Stargardt-like Clinical Characteristics and Disease Course Associated with Variants in the WDR19 Gene"

_genes, 2023, doi:10.3390/genes14020291_

Round 1
Reviewer 1 Report
Stargardt disease is common form of genetically inherited retinal disease affecting 30,000 people worldwide. It usually affects young children and symptoms get worse with age. Sajovic et al., in their paper are describing a patient carrying a novel mutation in the WDR19 gene. WDR19 has been linked to Stargardt disease in only a few studies earlier. The patient is this study is a double variant of the WDR19 gene, with a missense mutation in addition to a deletion mutation in exon 19 sequence that has not been studied before.
The paper is a good clinical paper that performs detailed comparison with patients known to be positive for Stargardt disease (i.e., carrying mutation in ABC4 gene; null or otherwise). The authors have used advanced imaging techniques that show the characteristic presence of Stargard-like phenotype in OCT and FAF imaging in their patient. The authors of the paper caution the readers that the phenotypic analysis is based one patient and that the phenotype has some deviation from the classical Stargardt-like phenotypes. As such the authors do acknowledge that this gene and its association with Stargadt disease needs further validation.
But because Stargardt disease is heterogenous in its phenotype, reporting mutation of this gene showing the disease symptom is important for future consideration while genetic testing patients.
I have the following question/comments for the authors.
1] Were the ABC4 patients tested for mutations in WDR19 gene?
2] In lines 157-159, can you be more descriptive about what PM2, PM4, and PP3 mean?
3] For Fig 5-9, the data can be made more visually clear if you can average the Y-axis values for each group and present as a bar graph.
Reviewer 2 Report
Gist/Summary: The authors work on a long followup of a (two decades) patient with ophthalmological description of a WDR19-Stargardt patient. the author sperformed detailed imaging analyses with detailed udnerstanding by comparing the WDR19 phenotype with equally well documented cohort of patients with ABCA4-Stargardt disease.
The manuscript is documented well but the authors could hav cited one of the well known articles in this field: https://www.ncbi.nlm.nih.gov/pmc/articles/PMC5270621/
(The Lines 148-150 could be moved up) Ethics and informed consent statement must be included in materials and methods as well
What was the whole exome sequencing chemistry the author used? IF so was it I5' UTRs + NCV regions? Pl elaborate
Any phenocopies associated with CNVs?
The WDR19 mutations containing rsids ( in case of existing or well known variant) and ss ids ( in cas eof novel/reported by the authors) may be added
were the two VoUS identified significant in other (European databases) sub population apart from GNOMAD?
Minor but essential:
Line 115: dilation pl correct it across all "pupil dilatation"
Lines 83/121: fluorescence mis-spelt
L126: represented A healthy
:158: Pl replace nor to OR
L237: had A relatively
L244: WDR19 patientS
L246: ARE known
L417: heterogeneous
The authors could knit small paras together
Scores on a scale of 0-5 with 5 being the best
Language: 4
Novelty: 4
Brevity: 3.5
scope/Relevance: 4
